# Fiber Metabolism, Procollagen and Collagen Type III Immunoreactivity in Broiler Pectoralis Major Affected by Muscle Abnormalities

**DOI:** 10.3390/ani10061081

**Published:** 2020-06-23

**Authors:** Maurizio Mazzoni, Francesca Soglia, Massimiliano Petracci, Federico Sirri, Giulia Lattanzio, Paolo Clavenzani

**Affiliations:** 1Department of Veterinary Medical Sciences, Alma Mater Studiorum–University of Bologna, 40064 Ozzano Emilia (B.O.), Italy; giulia.lattanzio2@unibo.it (G.L.); paolo.clavenzani@unibo.it (P.C.); 2Department of Agricultural and Food Sciences, Alma Mater Studiorum–University of Bologna, 47521 Cesena (F.C.), Italy; francesca.soglia2@unibo.it (F.S.); m.petracci@unibo.it (M.P.); federico.sirri@unibo.it (F.S.)

**Keywords:** chicken, muscle abnormalities, NADH-TR, αGPD, procollagen and collagen type III

## Abstract

**Simple Summary:**

The impressive production performances achieved by the modern chicken hybrids selected for meat production have indirectly predisposed the pectoral muscle to the onset and progression of abnormalities (i.e., white striping, wooden breast, and spaghetti meat). These myopathies affect the pectoralis major, with a high incidence rate and result in relevant economic losses for the poultry industry due to downgrading of the affected meat. These muscular abnormalities have been often associated with changes in ether the metabolism of the fibers (i.e., a shift from type IIB towards type IIA fibers) or to an increased deposition of collagen up to fibrosis. As collagen type III is typically observed in regenerating muscles, this and its precursor (procollagen type III) might play a role in the cellular processes, resulting in the development of white striping, wooden breast, and spaghetti meat abnormalities. According to our findings, both morphology and metabolism of the fibers were remarkably affected by the occurrence muscular abnormalities that are also associated with a profound modification in the connective tissue architecture. Intriguingly, an altered metabolism and an evident difference in the presence and distribution of procollagen and collagen type III was even observed in pectoralis major muscle from cases classified as unaffected.

**Abstract:**

The present study aimed to evaluate the muscle fiber metabolism and assess the presence and distribution of both procollagen and collagen type III in pectoralis major muscles affected by white striping (WS), wooden breast (WB), and spaghetti meat (SM), as well as in those with macroscopically normal appearance (NORM). For this purpose, 20 pectoralis major muscles (five per group) were selected from the same flock of fast-growing broilers (Ross 308, males, 45-days-old, 3.0 kg live weight) and were used for histochemical (nicotinamide adenine dinucleotide tetrazolium reductase (NADH-TR) and alpha-glycerophosphate dehydrogenase (α-GPD)) and immunohistochemical (procollagen and collagen type III) analyses. When compared to NORM, we found an increased proportion (*p* < 0.001) of fibers positively stained to NADH-TR in myopathic muscles along with a relevant decrease (*p* < 0.001) in the percentage of those exhibiting a positive reaction to α-GPD. In addition, an increased proportion of fibers exhibiting a positive reaction to both stainings was observed in SM, in comparison with NORM (14.3 vs. 7.2%; *p* < 0.001). After reacting to NADH-TR, SM exhibited the lowest (*p* < 0.001) cross-sectional area (CSA) of the fibers (−12% with respect to NORM). On the other hand, after reacting to α-GPD, the CSA of WS was found to be significantly larger (+10%) in comparison with NORM (7480 vs. 6776 µm^2^; *p* < 0.05). A profound modification of the connective tissue architecture involving a different presence and distribution of procollagen and collagen type III was observed. Intriguingly, an altered metabolism and differences in the presence and distribution of procollagen and collagen type III were even observed in pectoralis major muscle classified as NORM.

## 1. Introduction

The selection programs carried out in the past 50 years aiming at developing high growth-rate and breast-yield chicken hybrids have allowed a remarkable improvement in production profitability by increasing breast muscle yield. However, this pressure induced by the selection practices profoundly affected the structural architecture of the pectoralis major (PM) muscle (increased myofiber cross-sections and density, reduction in capillary density and capillary-to-fiber ratio, etc.) and, indirectly, resulted in the development of growth-related muscular abnormalities, namely, white striping (WS), wooden breast (WB), and spaghetti meat (SM) [1,2,3,4]. With regard to the impact of the selection practices, contrasting results were obtained in previous studies carried out to evaluate the fiber type composition of chicken PM belonging to different genotypes [5,6,7]. However, overall, a conversion from type I and IIA to type IIB fibers, and a consequent shift of the energetic metabolism towards the glycolytic pathway, has been associated with the selection for increased growth-rate in different species including chickens, pigs, and double-muscled cattle [5,8].

As for connective tissue, the skeletal muscles are composed of type I and III collagen. In detail, being mainly related to the embryonic development of the muscle, type III collagen is often associated with type I and plays a key role in its fibrillogenesis [9]. A switch from the embryonic type III to type I collagen is commonly observed during animal growth [3,9] and, at slaughtering age, broiler PM are mainly composed of type I collagen [3]. Conflicting results have been found concerning the relationship between collagen content and meat texture. However, an increased deposition of collagen type III within the endomysial and perimysial compartments was found to be positively correlated with muscle toughness [10], and an altered deposition of this collagen type has been associated with the occurrence of muscular dystrophies and fibrosis [11,12]. Thus, being an “embryonic” connective tissue that is typically observed in developing/regenerating muscles, collagen type III and its precursor (procollagen type III) might play a role in the cellular processes, resulting in the development of WS, WB, and SM abnormalities. Within this context, the present study aimed at evaluating the energetic metabolism of the fibers as well as assessing the distribution of procollagen and collagen type III within the endomysial and perimysial compartments of PM muscles affected by WS, WB, and SM conditions. In addition, since the PM classified as unaffected cases (NORM, without any gross superficial lesion) also exhibited altered histological features, the same analyses were carried out on PM belonging to a slow-growing genotype to evaluate the eventual differences related to the genetic background.

## 2. Materials and Methods

Twenty PM muscles were collected 3 hours post-mortem from the same flock of fast-growing broilers farmed and slaughtered under commercial conditions (Ross 308, males, 45 day-olds slaughtered at 3.0 kg live weight). The PM (5 per group) were classified according to their macroscopic features as NORM, WS, WB, or SM following the criteria adopted in our previous studies [13]. In detail, in order to avoid any interference, we did not consider muscles that were concurrently affected by more than one defect (i.e., combined WS/WB and WS/SM), and only severe cases were sampled. In addition, 3 PM were collected from chickens belonging to a slow-growing genotype farmed and slaughtered under commercial conditions (160-day-old Leghorn cocks slaughtered at 2.5 kg of live weight). The findings obtained for these slow-growing birds were compared with those found for NORM to evaluate the eventual differences related to the genetic background of the birds (fast-growing broilers selected for meat production and slow-growing birds belonging to an egg-type genotype). Samples for histochemical and immunohistochemical analyses were excised from the superficial section of the cranial portion of the PM according to the procedure described in our previous study [13], quickly frozen in isopentane (cooled with liquid nitrogen), and stored at −80 °C until processing. Serial cross-sections (10 μm thick) were cut on a cryostat microtome at −20 °C and mounted on poly-L-lysine-coated glass slides (Sigma-Aldrich, St. Louis, MO, USA).

### 2.1. Histochemical Analyses (NADH-TR and α-GPD) and Morphometric Measurements

The oxidative metabolism of the fibers was assessed by reduced nicotinamide adenine dinucleotide tetrazolium reductase (NADH-TR) staining [14], whereas the alpha-glycerophosphate dehydrogenase (α-GPD) method was used to evaluate the glycolytic metabolism according to the procedure described by Ishimoto et al. [15].

Morphometric evaluations were performed with a 10× objective lens using a Zeiss Axioplan microscope (Carl Zeiss, Oberkochen, Germany). The cross-sectional area (CSA) of the fibers was estimated by outlining their profile on the monitor screen using a computer mouse. At least 150 fibers per sample were measured in blind by two investigators. Every field measured about 705 × 10^3^ µm^2^. The CSA of the fibers (µm^2^) as well as the percentage of fibers positively stained to NADH-TR and/or α-GPD were assessed.

### 2.2. Procollagen and Collagen Type III

For immunohistochemistry, the avidin-biotin-peroxidase complex method was used. Briefly, cryostat sections were treated in 1% aqueous hydrogen peroxide solution for 30 min at room temperature (RT) to block endogenous peroxidase activity. Subsequently, the sections were incubated in phosphate buffer saline (PBS) containing 10% appropriate normal serum (goat and horse normal serum) in a humid chamber for 30 min at RT. The sections were then incubated overnight at 4 °C with the following primary antibodies: goat anti-procollagen type III diluted 1:1000 (Pro-COL3A1, sc-8779, Santa Cruz, CA, USA) and rabbit anti-collagen type III 1:1000 (ab185659, Abcam, Cambridge, UK). After washing, the sections were incubated at RT for 1 h with biotinylated goat anti-rabbit immunoglobulin G (IgG) and horse anti-goat IgG antibodies (Vector, Vector Laboratories, Burlingame, CA, USA), both diluted 1:200, and then treated with Avidin-Biotin Complex (ABC) (Vector elite kit, Vector Laboratories, Burlingame, CA, USA). The immune reactions were visualized through applying a 3,3’-diaminobenzidine (DAB) chromogen solution (Vector DAB kit, Vector Laboratories, Burlingame, CA, USA).

### 2.3. Antibody Specificity

The specificity of the goat anti-procollagen type III antibody was assessed by pre-adsorption with an excess of the homologous peptide (sc-8779 P, Santa Cruz, CA, USA), whereas the rabbit anti-collagen III is specific for the chicken collagen type III. Concerning the secondary antibodies, we performed further specificity tests by omitting the primary antibodies.

### 2.4. Statistical Analysis

Data concerning CSA of the fibers were analyzed through the one-way ANOVA option of Statistica 10 (StatSoft Inc., 2014, Tulsa, OK, USA) considering the occurrence and the type of abnormality (NORM, WS, WB, and SM) as a main effect. The same model was adopted to evaluate the results concerning NORM muscles and those belonging to their slow-growing counterpart. The same model was applied to analyze the data concerning fiber typing. However, the percentages of fibers positively stained to NADH-TR and/or α-GPD were subjected to arc sin transformation prior to analysis. Means were subsequently separated by using the parametric Tukey-honestly significant difference (HSD) test. All statistical differences were considered significant at a confidence level of 95% (*p* < 0.05).

## 3. Results

### 3.1. Histochemical Analyses (NADH-TR and α-GPD) and Morphometric Measurements

The results of the histochemical staining performed to identify the fibers having oxidative (positive to NADH-TR) and glycolytic (positive to α-GPD) metabolism are reported in Table 1, whereas images representative of the histochemical reactions are shown in Figure 1. Overall, the occurrence of muscular abnormalities significantly affected the proportion of fibers with glycolytic and oxidative metabolism. Indeed, if compared with NORM, a significant increase in the proportion of fibers positively stained to NADH-TR was found in WS- and SM-affected muscles (9.5 vs. 13.4 and 19.7%, respectively; *p* < 0.001), with the latter exhibiting the highest values. On the other hand, the percentage of fibers showing a positive reaction in WB (10.9%) did not significantly differ from that observed in unaffected cases (*p* > 0.05). Concurrently, a relevant increase in the proportion of fibers exhibiting a positive reaction to both NADH-TR and α-GPD was observed in SM-affected muscles, in comparison with NORM (14.3 vs. 7.2%; *p* < 0.001). Indeed, in NORM, several fibers were found to be positively stained to both NADH-TR and α-GPD (Figure 1A1,B1) and a substantially altered energetic metabolism was evident in WS, WB, and SM (Figure 1A2–A4,B2–B4), as depicted by the increased number of fibers positively stained to both NADH-TR (with an intense reaction taking place within the sarcoplasm) and α-GPD. 

Astonishing results were found by comparing the fiber types in fast-growing NORM and in PM belonging to the slow-growing genotype. Indeed, if compared to their slow-growing counterpart, a significantly higher proportion of fibers positively stained to NADH-TR was found in NORM (2.0 vs. 9.5%; *p* < 0.001) along with a remarkably lower percentage of those exhibiting a positive reaction to α-GPD (97.3 vs. 91.1%; *p* < 0.001) as well as to both the histochemical staining (1.4 vs. 7.2%; *p* < 0.001) (Table 1).

The results concerning the morphometric measurements carried out to evaluate the CSA of the fibers after their reaction to NADH-TR and α-GPD are reported in Table 2. Within the same group, the difference in the average CSA of the fibers assessed after the histochemical reaction are ascribable to the staining procedures. Indeed, the last step of NADH-TR staining protocol required the dehydration in alcohol of the sections, thus resulting in muscle fibers having slightly smaller CSA, whereas α-GPD sections were covered without any dehydration step. Overall, the occurrence of muscular abnormalities significantly affected the CSA of the fibers measured after reacting to NADH-TR and α-GPD, although inconsistent results, mainly ascribable to the staining protocols, were found. Indeed, by considering the results obtained after reacting to NADH-TR, if compared to the other groups, SM exhibited a significantly (*p* < 0.001) reduced CSA of the fibers (−12% in respect to NORM). On the other hand, after reacting to α-GPD, the CSA of WS and SM was found significantly larger (+10%) in comparison with NORM (7480 and 7339 vs. 6776 µm^2^; *p* < 0.05), whereas WB exhibited intermediate values (6978 µm^2^). Similarly, the CSA measured in PM belonging to the slow-growing genotype after reacting to NADH-TR and α-GPD significantly differed from their fast-growing counterpart. Indeed, if compared to those observed in the slow-growing genotype, a 41 and 34% increase in the CSA of the fibers was found in NORM muscles after reacting to NADH-TR (3740 vs. 6289 µm^2^; *p* < 0.001) and α-GPD (4499 vs. 6776 µm^2^; *p* < 0.001), respectively.

### 3.2. Procollagen and Collagen Type III

The expression and distribution patterns of procollagen and collagen type III are shown in Figure 2. Overall, a very heterogeneous staining with a broad range of staining intensity was observed for both procollagen and collagen type III in PM belonging to the fast-growing genotype (either NORM or affected by muscular abnormalities). An altered positivity of both procollagen and collagen III was observed in NORM—procollagen immunoreactivity was restricted to some areas of the perimysial spaces (Figure 2A1), whereas the staining for collagen III was very evident (Figure 2B1). In detail, endomysial collagen III was diffusely marked, while intensely stained areas were intermingled to weakly labelled ones in that composing the perimysial spaces. Similarly, an intensely labelled connective tissue was observed within the muscles affected by WS (Figure 2B2). A deeply altered muscle architecture was found in WS, in which part of the fibers was replaced by the proliferation of poorly organized connective tissue that was immunoreactive to both procollagen and collagen type III (Figure 2A2,B2). As for WB, a weak immunoreaction for procollagen type III was observed in both the endomysial and perimysial spaces. Furthermore, the immunohistochemical reaction for collagen type III revealed unexpected results—the connective tissue composing the endomysial septa was stained, whereas that composing the perimysial compartments reacted weakly and exhibited a poorly organized structure in which intensely stained collagen bundles were combined to faintly labelled ones (Figure 2A3,B3). In SM, a progressive rarefaction of the endomysial and perimysial connective tissue, leading to muscle fibers detaching from each other, was observed. An almost absent immunoreactivity to procollagen was evidenced and only the fibroblasts appeared stained. In addition, the defective endomysial connective tissue showed a strong collagen type III labelling (Figure 2A4,B4). 

The histochemical observation performed on the PM muscles belonging to the slow-growing genotype evidenced a regular architecture of the endomysial and perimysial connective tissue. Notwithstanding, procollagen around and intermingled into the muscle fibers was found to be weakly immunoreactive, whereas fibroblasts were intensely stained (Figure 2A). The architecture of the connective tissue appeared to be normally arranged, and collagen III was intensely marked (Figure 2B). 

## 4. Discussion

The fibers composing the skeletal muscle are commonly classified according to their contractile properties in mammals [16] and to their histochemical and histological characteristics in chickens [6]. In detail, in avian, the different fiber types are normally distinguished by combining the results obtained after reacting to both NADH-TR and/or acid/alkaline ATPase. In addition, α-GPD has also been previously used to assess the glycolytic metabolism of the skeletal muscle in rat, mouse, rabbit, monkey, and human [17,18,19]. On the basis of these traits, muscle fibers in chicken skeletal muscle can be classified as type I (slow oxidative), type IIA (fast oxido-glycolytic), and type IIX and IIB (fast glycolytic) [20]. 

Overall, the findings of the present study evidence that the occurrence of muscular abnormalities significantly affect the energetic metabolism of the fibers. Among the PM belonging to the fast-growing genotype, if compared with NORM, the significant reduction in the proportion of fibers having glycolytic metabolism (positively stained to α-GPD) observed in myopathic muscles along with an increased percentage of those displaying oxidative metabolism (positively stained to NADH-TR) is in agreement with a previous study in which an increased proportion of type I fibers was found in dystrophic muscles [21]. Within this context, it should be pointed out that previous studies carried out on mice [22] and humans [23] have demonstrated that type I fibers are less prone to develop dystrophic conditions. Indeed, in their early stage, dystrophies primarily affect type IIB fibers, whereas those having oxidative metabolism are involved only as the dystrophic condition get worse [24]. Therefore, the conversion from type IIB to type I fibers (as depicted by the increased proportion of fibers positively stained to NADH-TR and by the concurrent reduction in those exhibiting a positive reaction to α-GPD) may be considered as a compensatory mechanism to alleviate the symptoms and slow down the progression of the diseases, as previously hypothesized by Glaser and Masatoshi [25]. In agreement with this, the occurrence of a fiber type switching from type IIB to type I has been hypothesized to take place in WB and WS/WB-affected muscles [26,27] in which an upregulation of genes potentially indicating fiber-type switching following myofiber damage was observed. 

The significant differences in the proportion of fibers positively stained to either NADH-TR or α-GPD found by comparing NORM with their slow-growing counterpart support the preponderant presence of type IIB fibers within the PM muscles that, aside from their genetic background (either fast- or slow-growing genotypes), are predominantly composed of type IIB fibers. Thus, in agreement with Remignon et al. [28], it may be hypothesized that the fiber type composition is mainly related to the muscle function rather than to its proportion and growth-rate. However, it should be mentioned that, because of their different genetic backgrounds and growth curves, the PM belonging to the fast-growing hybrid and their slow-growing counterpart were sampled on the basis of the live weight of the animals that, therefore, were at different stages of their development, and this may have an influence on the fiber type composition of the muscle tissue itself. In addition, although contrasting results were observed in previous studies in which the fiber type was assessed in PM belonging to chickens having different genetic background [5,6,7], a conversion from type IIA to IIB was found in association with the selection for an increased muscularity in chickens, pigs, and double-muscled cattle [5,8,29].

It should be also pointed out the presence of aberrant fibers positively stained to both NADH-TR and α-GPD in PM belonging to the fast-growing hybrid (either in WS-, WB-, or SM-affected muscles and even in those classified as NORM). The presence of these abnormal fibers (having a small or large caliber) might be explained by considering the remarkable pressure induced by the selection programs with the aim to develop high growth-rate and breast-yield hybrids. Indeed, fibers positively stained to both NADH-TR and α-GPD and exhibiting large CSA (and rounded profile) are likely hypercontracted fibers and may be considered as an attempt of the muscle tissue to overcome the degenerative processes, activating both the oxidative and glycolytic metabolism to produce energy. On the other hand, those positively stained to both NADH-TR and α-GPD and with a small caliber might result from the regenerative processes taking place within the PM—these fibers are likely fixed at an intermediate stage of their conversion from type I to type IIB [30]. Indeed, although the PM is predominantly composed of oxidative fibers in embryonic stages, these are converted to type IIB within 2 to 3 weeks post-hatch [31]. However, since this conversion has been demonstrated to remain incomplete within the PM of dystrophic chicks [31], a similar mechanism might be hypothesized to take place in PM belonging to the fast-growing genotype (whose microscopic traits can be comparable with those typically found in muscle dystrophies). 

Intriguingly, in PM belonging to the fast-growing genotype (WS, WB, and SM, as well as those classified as NORM cases), we found the presence of fibers exhibiting an intense staining to NADH-TR, whereas these were not observed in PM belonging to the slow-growing genotype. NADH-TR staining is based on the development of a purple-blue formazan pigment that is able to mark the site of NADH enzyme activity, the mitochondria. Thus, the intense NADH-TR reaction observed in PM belonging to fast-growing hybrids may reflect the development of mitochondrial disorders that may arise as a primary dysfunction of the mitochondrial respiratory chain [32]. In agreement with this, in a recent study carried out on human skeletal muscles affected by mitochondrial disorders, fibers exhibiting a subsarcolemmal accumulation of these organelles similarly exhibited an intense staining to NADH-TR [33].

As for the CSA, the findings obtained in the present study may be at least partly explained by considering the different proportion of fibers with oxidative and glycolytic metabolism. Indeed, the 12% reduction in CSA observed in SM after staining for NADH-TR in comparison with NORM may be ascribed to the increased proportion (above twofold) of fibers with oxidative metabolism, and reduced caliber if compared to the glycolytic forms found in these muscles. On the other hand, when compared to NORM, the 10% increase in the CSA of the fibers observed in WS and SM after reacting to α-GPD may be explained by considering the intense regenerative processes taking place in these muscles. Indeed, although the occurrence of muscle regeneration has been demonstrated to occur in WS, WB, and SM muscles, a different progression of this condition has been recently hypothesized, with WS being considered at a later stage of regeneration when compared to WB [13]. The findings of the present study seem to support this hypothesis—WS, seems to be in an advanced step of regeneration in which the diameter of the regenerated fibers is similar to that of the pre-existing ones and thus larger. On the other hand, WB muscles can be considered in an earlier stage of regeneration in which, as previously observed by Velleman and Clark [34], the diameter of the fiber is slightly smaller, also as a consequence of the increased expression of transforming growth factor beta (TGF-β) and myostatin, inhibitors of cell proliferation and differentiation. 

The comparison of CSA assessed in NORM with the values measured in PM belonging to the slow-growing genotype revealed remarkable differences that were mainly related to the hypertrophic growth of the fibers induced by the selection practices. Indeed, the findings of the present study are in agreement with those of previous research in which, when compared with those belonging to local breeds and egg-type genotypes, a remarkably increased CSA of the fibers was observed in fast-growing birds [1,3].

Collagen is the major component of the connective tissue and plays an essential role in differentiating and maintaining the structure of the developing and mature muscle fibers, respectively [35]. In this regard, collagen type III is mainly found in association to type I and plays a key role in its fibrillogenesis [9]. Indeed, because of its inherent structure, collagen type III may contribute to the extensibility and elasticity of the connective tissue [36], and its altered deposition has been previously associated with various muscular dystrophies and fibrosis [11,12]. Overall, the heterogeneous staining to procollagen and collagen type III observed in the fast-growing hybrid (aside from the type of abnormality) might be also explained by the eventual changes in amino acid composition previously observed in collagen isolated from dystrophic chickens [11]. Indeed, a selective removal of the polar side-chains and their replacement with non-polar amino acids was observed and resulted in a weakened helical structure of the collagen fibrils, altering both the functional and structural traits of the molecules [11]. With regard to WS, the immunoreactivity for procollagen and collagen type III observed in these muscles suggests an intense collagen synthesis taking place. Similar results were previously reported in dystrophic chickens in which an increased amount of collagen type III was found and attributed to an altered synthesis [37]. Concurrently, the presence of more immature and newly synthesized collagen fibrils was observed by these authors, suggesting that the fibrotic processes might not be a compensatory replacement of the necrotic fibers, but instead may be an active process involving an increased deposition of connective tissue and a concomitant decrease in muscle protein synthesis [37]. On the other hand, the weak positivity to both procollagen and collagen type III observed in WB (in which intensely stained endomysial spaces were intermingled to faintly labelled perimysial compartments) is in agreement with previous immunohistochemical features observed in double-muscled cattle in which the perimysial collagen surrounding the developing muscles were weakly stained [38]. As for SM, the progressive rarefaction of the perimysial connective tissue may account for the distinctive phenotype associated with this condition in which the detachment of the fiber bundles composing the PM has been observed [39], suggesting an altered collagen turnover and synthesis, thus resulting in an altered immunoreactivity to procollagen type III. When compared to NORM, the immunohistochemical observations carried out on PM belonging to the slow-growing genotype evidenced an orderly organized connective tissue composing both the endomysial and perimysial septa. 

## 5. Conclusions

In conclusion, both the morphology and the metabolism of the fibers were found to be remarkably affected by the occurrence of WS, WB, and SM abnormalities that are associated with a profound modification of the connective tissue architecture. Intriguingly, an altered metabolism and an evident difference in the presence and distribution of procollagen and collagen type III was observed by microscopically examining the PM classified as NORM. The outcomes of the present study support the argumentation of a recent review paper [40] in which the potential limits of selection in poultry were deeply examined. In detail, as discussed by Tixier-Boichard [40], although it seems possible to obtain a further improvement in the production performances of the animals, eventual unfavorable correlated effects (i.e., the occurrence of muscular abnormalities) need to be carefully considered.

## Figures and Tables

**Figure 1 animals-10-01081-f001:**
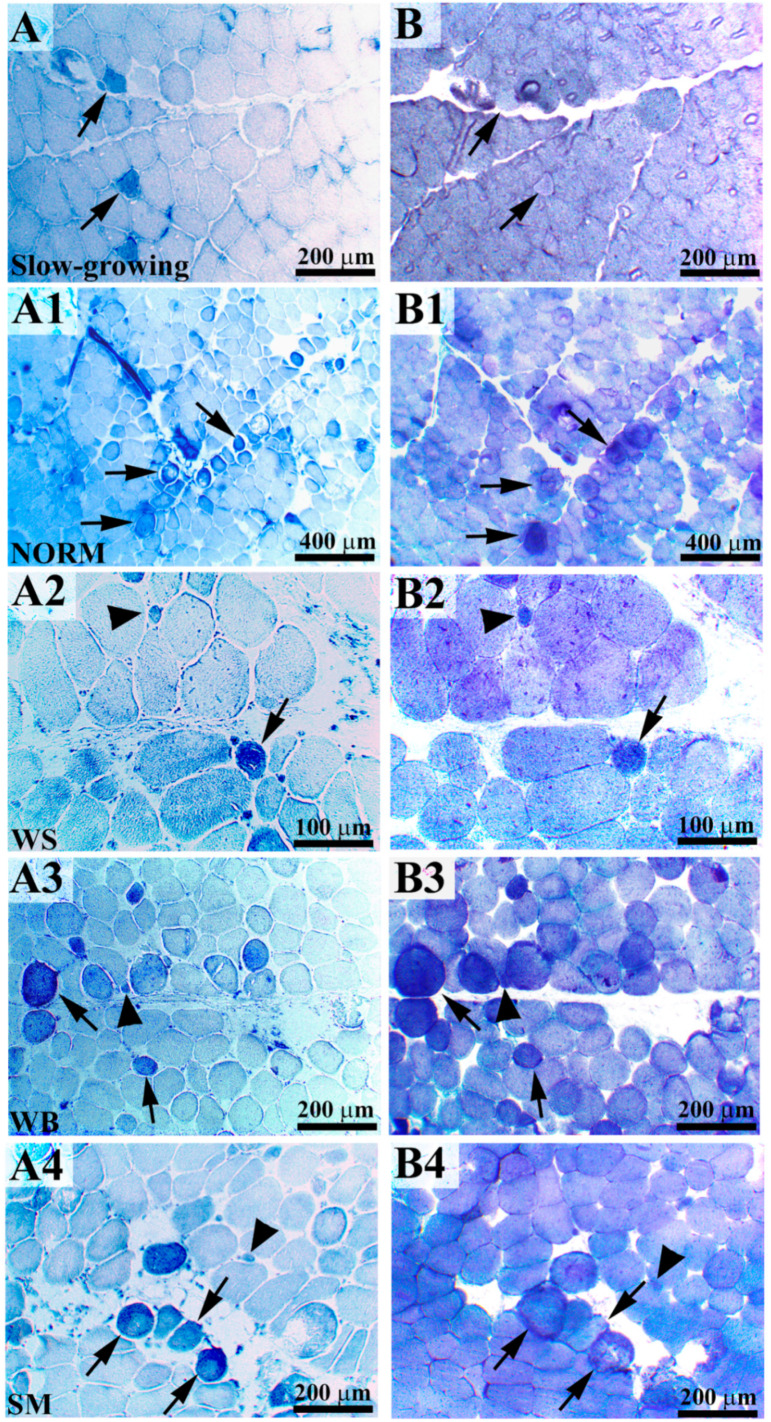
Serial cross-sections of pectoralis major (PM) belonging to a slow-growing genotype (**A**,**B**) are reported in comparison with the normal genotype (NORM) (**A**1,**B**1)). Few nicotinamide adenine dinucleotide tetrazolium reductase (NADH-TR)-positive fibers were detected within the pectoralis major of the slow-growing genotype ((**A**), arrows) whereas an intense and diffuse positivity for alpha-glycerophosphate dehydrogenase (α-GPD) activity was found ((**B**), arrows). In NORM, some fibers exhibiting an intense oxidative and glycolytic metabolism ((**A**1,**B**1), arrows) were found. In white striping (WS)-, wooden breast (WB)-, and spaghetti meat (SM)-affected muscles, many fibers exhibited an intense reaction to NADH-TR ((**A**2–**A**4), arrows) and α-GPD ((**B**2–**B**4), arrows). Within these muscles, we observed intensely stained, positive small-caliber regenerative fibers in correspondence with larger-caliber fibers ((**A**2–**A**4,**B**2–**B**4), arrowheads).

**Figure 2 animals-10-01081-f002:**
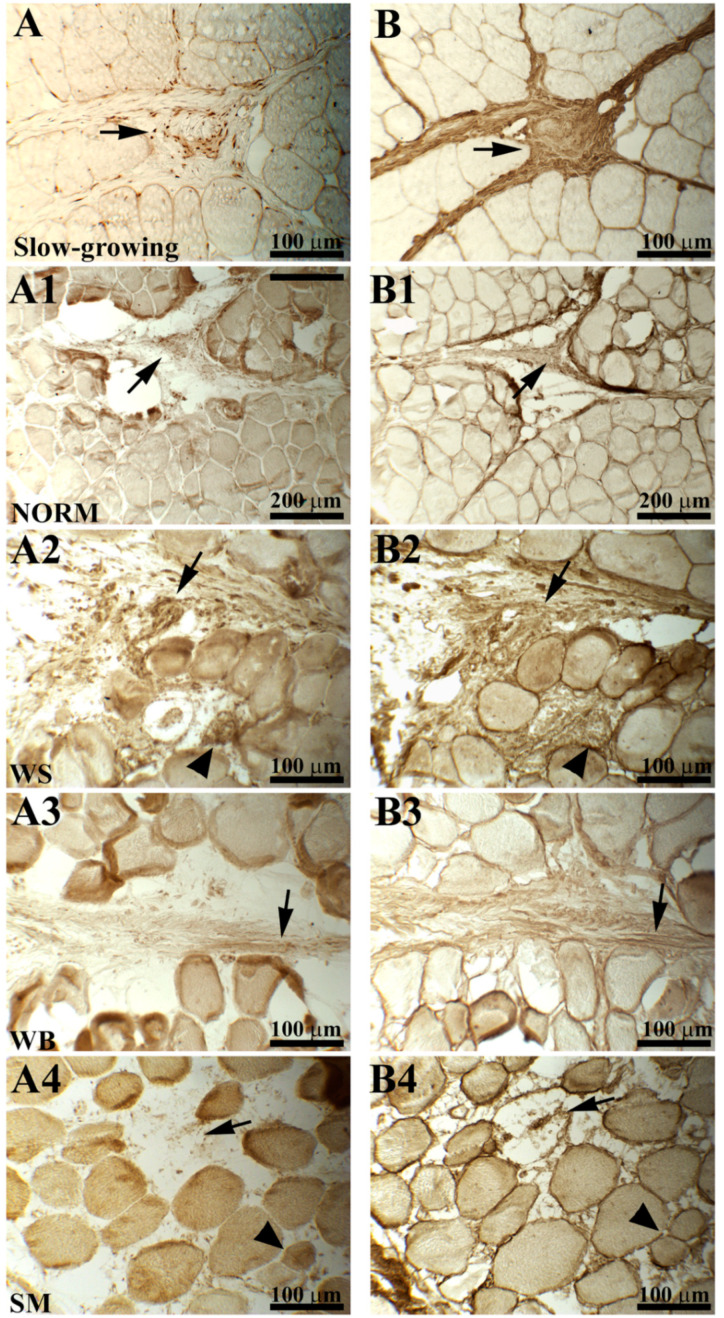
Serial cross-sections of chicken pectoralis major (PM) muscles belonging to a fast-growing genotype classified as normal cases (NORM) and those affected by white striping (WS), wooden breast (WB), or spaghetti meat (SM) abnormalities showing the immunoreaction to procollagen type III (**A–A4**) and collagen type III (**B–B4**) antibodies. Immunoreactivity to procollagen type III was almost absent in PM belonging to the slow-growing genotype, and only the fibroblasts were intensely stained ((A), arrow). Collagen type III was intensely marked and appeared normally arranged within the endomysial and perimysial compartments ((B), arrow). In NORM, immunoreactivity to procollagen was restricted to some areas of the perimysial spaces ((A1), arrow). In addition, collagen type III within the endomysial compartment was intensely labelled, whereas in the perimysial septa, intensely stained areas were intermingled to weakly marked ones ((B1), arrow). A profoundly altered muscle architecture was found in WS- and WB-affected muscle (A2,A3,B2,B3)—degenerated muscle fibers were replaced by poorly organized connective tissue immunoreactive to both procollagen and collagen type III ((A2,B2), arrowheads). The diffused connective tissue observed in (A2) and (B2) (WS muscles) was intensely stained (arrows), whereas weak immunoreactions were observed in WB-affected muscles (A3,B3) in which intensely stained collagen bundles alternated to faintly labelled ones (arrows). An evident detachment of the muscle fibers composing the PM was observed within the SM-affected muscles (A4,B4). Immunoreaction to procollagen was almost absent (only the fibroblasts were stained), while the endomysial connective tissue exhibited a strong collagen III labelling.

**Table 1 animals-10-01081-t001:** Number of fibers (expressed as percentage) positively stained to reduced nicotinamide adenine dinucleotide tetrazolium reductase (NADH-TR) and/or alpha-glycerophosphate dehydrogenase (α-GPD) in pectoralis major muscles belonging to a fast-growing genotype exhibiting either macroscopically normal appearance (NORM) or affected by white striping (WS), wooden breast (WB), or spaghetti meat (SM) abnormalities. The results obtained in fast-growing NORM muscles were also compared with those found in pectoralis major belonging to a slow-growing genotype. Values are expressed as mean ± standard error of the mean (SEM).

	+NADH-TR (%)	+α-GPD (%)	+NADH-TR/+ α-GPD (%)
NORM	9.5 ^c^ ± 0.3	91.1 ^a^ ± 0.3	7.2 ^b^ ± 0.5
WS	13.4 ^b^ ± 0.3	87.7 ^b^ ± 0.3	9.6 ^b^ ± 0.3
WB	10.9 ^bc^ ± 0.5	89.9 ^ab^ ± 0.3	8.0 ^b^ ± 0.8
SM	19.7 ^a^ ± 3.2	82.3 ^c^ ± 0.4	14.3 ^a^ ± 3.6
*p*-value	<0.001	<0.001	<0.001
Fast-growing (NORM)	9.5 ^a^ ± 0.3	91.1 ^b^ ± 0.3	7.2 ^a^ ± 0.5
Slow-growing	2.0 ^b^ ± 0.5	97.3 ^a^ ± 0.2	1.4 ^b^ ± 0.4
*p*-value	<0.001	<0.001	<0.001

^a–c^ = *p* < 0.05. Mean values followed by different superscript letters significantly differ among the groups.

**Table 2 animals-10-01081-t002:** Cross-sectional area (CSA) of the fibers stained for reduced nicotinamide adenine dinucleotide tetrazolium reductase (NADH-TR) or alpha-glycerophosphate dehydrogenase (α-GPD) in pectoralis major muscles belonging to a fast-growing genotype exhibiting either macroscopically normal appearance (NORM) or affected by white striping (WS), wooden breast (WB), or spaghetti meat (SM) abnormalities. The results obtained in fast-growing NORM muscles were also compared with those found in pectoralis major belonging to a slow-growing genotype. Values are expressed as mean ± SEM.

	NADH-TR (µm^2^)	α-GPD (µm^2^)
NORM	6289 ^a^ ± 201	6776 ^b^ ± 170
WS	6460 ^a^ ± 207	7480 ^a^ ± 161
WB	6639 ^a^ ± 219	6978 ^ab^ ± 229
SM	5544 ^b^ ± 158	7339 ^a^ ± 171
*p*-value	<0.001	<0.05
Fast-growing (NORM)	6289 ^a^ ± 201	6776 ^a^ ± 170
Slow-growing	3740 ^b^ ± 87	4499 ^b^ ± 90
*p*-value	<0.001	<0.001

^a,b^ = *p* < 0.05. Mean values followed by different superscript letters significantly differ among the groups.

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
