# Peer review of "Fiber Metabolism, Procollagen and Collagen Type III Immunoreactivity in Broiler Pectoralis Major Affected by Muscle Abnormalities"

_animals, 2020, doi:10.3390/ani10061081_

Round 1

Reviewer 1 Report

The paper presented for review brings new elements to the current state of knowledge regarding muscle fiber metabolism and the presence and distribution of procollagen and collagen type III in Pectoralis major muscles in broiler chickens affected by abnormalities meat: white striping (WS), wooden breast (WB) and spaghetti meat ( MS).

The purpose of the work is clearly stated. The conclusions of the conducted research are clear and result from the obtained research results.

The material used for the tests is sufficient, the test methods have been selected accordingly.

The table layout is correct. The differences between the groups were marked correctly.

Photographic documentation and its description is very accurate.

Discussing the results against the background of other authors is very detailed.

The publications cited by the authors of the article are well selected. For the most part, the authors refer to the latest knowledge published in renowned scientific journals.

Author Response

Reviewer 1

The paper presented for review brings new elements to the current state of knowledge regarding muscle fiber metabolism and the presence and distribution of procollagen and collagen type III in Pectoralis major muscles in broiler chickens affected by abnormalities meat: white striping (WS), wooden breast (WB) and spaghetti meat (MS).

The purpose of the work is clearly stated. The conclusions of the conducted research are clear and result from the obtained research results.

The material used for the tests is sufficient, the test methods have been selected accordingly.

The table layout is correct. The differences between the groups were marked correctly.

Photographic documentation and its description is very accurate.

Discussing the results against the background of other authors is very detailed.

The publications cited by the authors of the article are well selected. For the most part, the authors refer to the latest knowledge published in renowned scientific journals.

Reviewer 2 Report

The authors have a good understanding of the subject and has covered all relevant aspects.

My comments are minor and may be considered in any revision.

33    NORM (7,480 vs. 6,776 μm2; p < 0.05) should be 7.480 vs. 6.776

97     0,705 should be 0.705

110   3,3 should be 3.3

176  7,479.9 Vs. 6,776.4  should be 7.479.9 Vs. 6.776.4

177   6,978.3 and 7,338.7 should be  6.978.3 and 7.338.7

181    3,739.5 vs. 6,288.5 should be 3.739.5 vs. 6.288.5  (4,498.8 vs. 6,776.4…) should be 4.498.8 vs. 6.776.4

183 - In title  is parenthesis – where is the end of the parenthesis?

Table 2. Cross Sectional Area (CSA of the fibers stained for NADH-TR or α-GPD in Pectoralis major muscles due to a fast-growing genotype exhibiting either macroscopically normal appearance (NORM) or affected by White Striping (WS), Wooden Breast (WB) and Spaghetti Meat (SM) abnormalities. The results obtained in fast-growing NORM muscles were also compared with those found in P. major belonging to a slow-growing genotype.

In Table 2  - All values (NORM, WS, WB and SM) must be with a FULL STOP!

444    

Glaser, J.; Masatoshi, S. Skeletal muscle fiber type in neuromuscular disease (chapter 5th). In Muscle Cells 442 and Tissue: Current Status of Research Field (Edited by K. Sakuma); Intech Open, London Bridge Street, 443 London, UK, 2018; pp. 65–79. 444   - 2018 – bold

Author Response

The authors have a good understanding of the subject and has covered all relevant aspects.

My comments are minor and may be considered in any revision.

33    NORM (7,480 vs. 6,776 μm2; p < 0.05) should be 7.480 vs. 6.776

We thank the reviewer for his/her comment. We have removed the comma indicating the thousands

97     0,705 should be 0.705

We replaced the comma with the period

110   3,3 should be 3.3

In the name of the organic compound 3,3’- diaminobenzidine chromogen solution, “3,3’ “ has not been modified because it’s related to its specific name.

176  7,479.9 Vs. 6,776.4  should be 7.479.9 Vs. 6.776.4

We thank the reviewer for his/her comment. We have removed the comma indicating the thousands.

177   6,978.3 and 7,338.7 should be  6.978.3 and 7.338.7

We thank the reviewer for his/her comment. We have removed the comma indicating the thousands

181    3,739.5 vs. 6,288.5 should be 3.739.5 vs. 6.288.5  (4,498.8 vs. 6,776.4…) should be 4.498.8 vs. 6.776.4

We thank the reviewer for his/her comment. We have removed the comma indicating the thousands

183 - In title is parenthesis – where is the end of the parenthesis?

Table 2. Cross Sectional Area (CSA of the fibers stained for NADH-TR or α-GPD in Pectoralis major muscles due to a fast-growing genotype exhibiting either macroscopically normal appearance (NORM) or affected by White Striping (WS), Wooden Breast (WB) and Spaghetti Meat (SM) abnormalities. The results obtained in fast-growing NORM muscles were also compared with those found in P. major belonging to a slow-growing genotype.

Following the suggestion, the end of the parenthesis has been placed as follow: “Cross Sectional Area (CSA) of the fibers..”

In Table 2  - All values (NORM, WS, WB and SM) must be with a FULL STOP!

We thank the reviewer for his/her comment. We have removed the comma indicating the thousands

444    Glaser, J.; Masatoshi, S. Skeletal muscle fiber type in neuromuscular disease (chapter 5th). In Muscle Cells 442 and Tissue: Current Status of Research Field (Edited by K. Sakuma); Intech Open, London Bridge Street, 443 London, UK, 2018; pp. 65–79. 444   - 2018 – bold

Following the suggestion from the reviewer, the reference has been properly modified ad follow:       29.            Glaser, J.; Masatoshi, S. Skeletal muscle fiber type in neuromuscular disease (chapter 5th). In Muscle Cells and Tissue: Current Status of Research Field (Edited by K. Sakuma); IntechOpen, London Bridge Street, London, UK, 2018; pp. 65–79.

Reviewer 3 Report

The article is well done and well explained.

Just the correction of a few spelling mistakes and a comment:

In general, the units of area should be represented as μm2 or mm2.µm2 or mm2 appears throughout the text. Please correct

Also, in lines 96 and 97, 705x103 should be written as 705x103. And the figures 0,705 should be written as 0.705 in the same lines.

Finally a comment. It seems strange that the percentage of fibers stained with α-GPD appears as no significant in Table 1. With the differences presented and the sem shown, they would be expected to be differences. This is just a comment, but I encourage authors to review the data and results to verify that nothing is wrong.

Author Response

The article is well done and well explained.

Just the correction of a few spelling mistakes and a comment:

In general, the units of area should be represented as μm2 or mm2.µm2 or mm2 appears throughout the text. Please correct

We thank the reviewer for his/her comment and following the suggestion we have reported the units related to the area of the fibers always using μm2 through the whole text.

Also, in lines 96 and 97, 705x103 should be written as 705x103. And the figures 0,705 should be written as 0.705 in the same lines.

Following the suggestion, we have reported the units related to the area of the fibers always using μm2 through the whole text and therefore we have deleted the area reported in mm2 (0.705).

Finally a comment. It seems strange that the percentage of fibers stained with α-GPD appears as no significant in Table 1. With the differences presented and the sem shown, they would be expected to be differences. This is just a comment, but I encourage authors to review the data and results to verify that nothing is wrong.

The results reported in Table 1 have been carefully checked and no differences have been found among the groups.